# Joint-outcome prediction markets for climate risks

**Mark S. Roulston** [1,2], **Kim Kaivanto**[2,3]*

1 Economics and Policy Institute, Land, Environment, University of Exeter, Exeter, United Kingdom,
2 CRUCIAL: Climate Risk and Uncertainty Collective Intelligence Aggregation Laboratory, United Kingdom,
3 Lancaster University Management School, Lancaster University, Lancaster, United Kingdom

* k.kaivanto@lancaster.ac.uk

## Abstract

Predicting future climate requires the integration of knowledge and expertise from a wide range of disciplines. Predictions must account for climate-change mitigation policies which may depend on climate predictions. This interdependency, or "circularity", means that climate predictions must be conditioned on emissions of greenhouse gases (GHGs). Long-range forecasts also suffer from information asymmetry because users cannot use track records to judge the skill of providers. The problems of aggregation, circularity, and information asymmetry can be addressed using prediction markets with joint-outcome spaces, allowing simultaneous forecasts of GHG concentrations and temperature. The viability of prediction markets with highly granular, joint-outcome spaces was tested with markets for monthly UK rainfall and temperature. The experiments demonstrate these markets can aggregate the judgments of experts with relevant expertise, and suggest similarly structured markets, with longer horizons, could provide a mechanism to produce credible forecasts of climate-related risks for policy making, planning, and risk disclosure.

## Introduction

Forecasting climate change is an important and multi-faceted prediction problem. Predicting climate *given* concentrations of greenhouse gases (GHGs) requires combining expertise in meteorology, oceanography, and other physical sciences; predicting what concentrations of GHGs might be in the future extends the problem into the realms of demographics, economics, policy, and technological innovation.

The cornerstone of climate forecasting is coupled general circulation models (CGCMs). These models simulate the dynamics and thermodynamics of the atmosphere and ocean. They are relatives of the numerical weather prediction models used to produce short to medium range (1 to 14 day ahead) weather forecasts. The main differences are that climate models include a dynamical ocean component, and they are run at a lower resolution because they are used to produce simulations lasting decades rather than weeks. State-of-the-art CGCMs, such as the U.K. Met Office's HadGEM3 models, have horizontal resolutions of approximately 60 km [1]. CGCMs require large amounts of computing power and historically have been the preserve of major institutions and government agencies. The Coupled Model Intercomparison

**Data Availability Statement:** https://dataverse.harvard.edu/dataset.xhtml?persistentId=doi:10.7910/DVN/797RXY.

**Funding:** The experimental markets were run while MR was employed by Winton Group. Winton also provided the cash incentives for institutions

participating in the markets. The manuscript was prepared after MR had left Winton and Winton did not play a role in the decision to publish, or in the preparation the manuscript.

**Competing interests:** The authors have declared that no competing interests exist.

Experiments (CMIP) run by the World Climate Research Programme is a framework to compare and understand differences between CGCMs. The most recent CMIP6 exercise involved 100 models from 50 different modelling centers [2]. Interpreting what the differences in model predictions from different CGCMs might be telling us about uncertainty in climate forecasts is complicated. Ensembles of simulations from different models cannot be interpreted as probabilistic forecasts, not least because the models are not independent as they share common "ancestors", or they have been influenced by the design of other models [3]. While much of what is understood about the climate system is codified in CGCMs, climate scientists also possess more tacit expertise about the strengths and weaknesses of different models and what aspects of their predictions are likely to be most reliable. Synthesizing the output of different models and combining it with the knowledge of experts in a structured and quantitative way is one of the challenges confronting the production of consensus forecasts for policymakers and planners.

One of the most important determinants of climate on a timescale of decades is the radiative forcing that results from GHGs in the atmosphere. As part of the CMIP experiments, models are run with specified concentration pathways of GHGs ranging from ones where the concentration of carbon dioxide ($CO_2$) in 2100 is similar to its present value (400 ppm) to scenarios where the concentration of $CO_2$ has increased to more than 1200 ppm. Concentrations of other GHGs (e.g., methane and nitrous oxides) are also specified in these scenarios. The amount of warming over the next century, according to CGCMs, is highly sensitive to GHG concentrations, ranging from 1.5˚C in the more benign scenarios to more than 4˚C should $CO_2$ exceed 1000 ppm. Climate forecasts are thus conditional on the GHG concentration scenario used by the models. Producing an unconditional forecast of future climate requires a view on the probability of different GHG concentration pathways.

The likelihood of different atmospheric concentrations of GHGs will be strongly affected by policies to reduce GHG emissions, and these policies will, at least partly, be influenced by predictions of future climate. In the context of inflation forecasting and interest rate setting this kind of interdependency has been called the "circularity problem" [4–6]. Circularity creates ambiguity in the interpretation of unconditional forecasts: if only modest warming is predicted over the next thirty years, this could be because forecasters do not believe the climate is sensitive to GHG concentrations, *or* it could be because they believe there will be dramatic reductions in GHG emissions. The former interpretation means little action is needed from policy makers whereas the latter interpretation implies a robust policy response is expected. Policy makers need to be able to differentiate these situations if the forecast is to be useful for informing policy. This issue is typically addressed by conditioning physical climate forecasts on scenarios representing different emissions pathways. The challenge of predicting the likelihoods of the different emissions scenarios is treated as a separate problem that, arguably, has not received the same attention as physical climate forecasting. For example, the likelihood of the RCP8.5 scenario from the Intergovernmental Panel on Climate Change (IPCC) is a matter of ongoing dispute among climate scientists [7–9].

As mentioned above, the resolution of CGCMs is typically around 60 km, but many decisions concerning adaptation and resilience to climate change require information on smaller scales. The process of producing higher resolution predictions from CGCM simulations is called downscaling. Several different approaches are used for downscaling. Regional coupled models (RCMs) simulate the dynamics and thermodynamics of the atmosphere at a higher resolution than CGCMs, but over a limited area [10]. They can be "nested" within a CGCM from which they get their boundary conditions. In contrast, statistical approaches use simulated quantities from a CGCM as predictor variables to predict quantities, such as temperature and rainfall, at higher spatial resolutions. Statistical approaches can include anything from

traditional linear modelling up to more sophisticated machine learning methods, such as neural networks [11]. What they have in common is that the models must be fitted or trained using CGCM simulations of past climates and observations at the required granularity. They typically make the assumption that the relationship between the larger scale climatic variables, directly modelled by the CGCMs, and more local climate, will remain the same, even in a changing climate.

In addition to public-sector CGCM modelling centers, there is now an increasing number of private sector companies—climate service providers (CSPs)—that produce high resolution climate forecasts of physical climate risks. Private sector companies have contributed to innovation in short-range weather forecasting for many years but there is an important difference between short-range forecasting and climate prediction: providers of short-range weather forecasts can establish track records of skill in a reasonable amount of time on which they can be judged. Users of long-range climate forecasts, however, cannot use historic track records to select a source of forward-looking climate information. The long "discovery time" associated with climate forecasts creates an information asymmetry between the user and the provider, who is likely to have a better understanding of their quality. This is known as the "lemon problem" in economics, a reference to the market for used cars [12]. When users cannot ascertain the quality of forecasts it is hard for them to place a value on the forecasts. In response, providers might be reticent to invest in improving forecast accuracy. Markets suffering from information asymmetry can thus experience a self-reinforcing decline in the average quality of products, culminating in market failure. The information asymmetry can be mitigated by sellers being held accountable for quality through voluntary warranties or statutory protections (colloquially known as "lemon laws" in the market for used cars). Agreeing to be paid based on accuracy allows providers to signal confidence in the quality of their forecasts. Paying forecasters according to a proper scoring rule can incentivize them to only make a prediction if they believe they have skill, even for a single forecast [13].

This trinity of problems affecting climate forecasts—distributed knowledge, circularity of predictions, and information asymmetry—could all be addressed using carefully designed "prediction markets" [14].

Prediction markets are designed to elicit and aggregate information, rather than transfer the ownership of assets or risk. They have been advocated as a way of aggregating information to produce collective forecasts, particularly when information and expertise is dispersed [15]. These markets are typically built around contingent contracts, known as "Arrow-Debreu" securities, that pay out 1.00 unit if a specified event occurs. These contracts are traded and, assuming the prevailing price (between 0.00 and 1.00) reflects the expected payout, it can be interpreted as a "market-based" probability of the specified event occurring. Prediction markets can continually assimilate new information as it becomes available to participants. Also, if the units represent real money, there is an incentive for participants to provide accurate and timely information, meaning that prediction markets can be an "incentive compatible" method for eliciting participants' true beliefs.

Prediction markets are neither a replacement, nor an alternative, to predictive models and expertise; they are a mechanism for synthesizing the outputs of different types of models and other forms of knowledge. Mechanisms to synthesize disparate approaches to forecasting are required more than ever due to the proliferation of new methods, such as machine learning, for climate prediction, as well as continued improvements to more traditional models. Prediction markets with expert participants have outperformed survey methods for predicting the reproducibility of scientific research [16] and they have been suggested as an appropriate tool to aggregate the diverse expertise required to make climate forecasts and to give those forecasts credibility [17–20]. Markets for year-ahead global temperature have been run on commercial

prediction market exchanges, such as Intrade and Smarkets. In the Intrade market, the global temperature anomaly for 2012 was partitioned into intervals of 0.05˚C and contingent contracts defined for each interval [21]. The Intrade platform used a *continuous double auction* which directly matches buyers and sellers of contracts. Unfortunately, due to the relatively large number of contract types, and the esoteric nature of the topic, some of the markets suffered from low levels of trading, making it difficult to infer reliable probabilities from prices. Such thinly traded markets are described as lacking liquidity. These markets are also "zero sum": the rewards of informed (or lucky) participants must come from the losses of uninformed (or unlucky) ones. If the market does not attract enough uninformed participants, the potential rewards are not great enough to attract informed ones.

Even if the liquidity problem can be solved, with longer-horizon temperature-only markets the problem of circularity makes their interpretation ambiguous. The circularity problem can be addressed by defining a two-dimensional space of possible outcomes. For example, GHG concentration (the policy variable) is partitioned into intervals on one axis and temperature anomaly is partitioned on the other. Therefore, each outcome is defined by simultaneous values of GHG concentration and temperature. This structure allows participants to take contingent positions that the temperature will lie within a certain range *if* the concentration of GHGs falls within a defined range. Each grid cell has a price (between 0.00 and 1.00) associated with it. Marginal price distributions for GHG concentration and temperature can be obtained by summing over each dimension separately, but the two-dimensional structure means that the market also implies a joint distribution of GHG concentration and temperature; essentially providing an estimate of transient climate sensitivity.

A problem with a joint-outcome space, however, is that it greatly increases the number of outcomes, and hence the number of contract types to be traded. This exacerbates the liquidity problem. It might be unlikely that any two participants will want to trade identical collections of contracts, complicating the task of directly matching buyers and sellers; this is known as the "double coincidence of wants".

The problems of matching buyers with sellers and low liquidity can be solved by using an automated market maker, an algorithm that will always buy or sell baskets of contracts. All trades are with the market maker that sets prices based on its relative exposure to different outcomes. Market makers exist in many financial markets, and bookmakers in traditional betting can also be thought of as market makers. The aim of traditional market makers, however, is to make a profit from the "spread", by selling contracts for slightly more than they buy them. But a market maker doesn't need to be motivated by profit. Instead, a market maker can be willing to lose money in return for good information. Such a subsidised market maker changes a zero-sum market into a positive-sum market, essentially taking the place of uninformed participants so that informed participants always have an incentive to take part. When subsidising a market through a market maker it must be carefully designed so that subsidies incentivise the provision of accurate information and cannot be "gamed" by participants. The Logarithmic Market Scoring Rule (LMSR) market maker [22] rewards participants according to the logarithmic scoring rule—a well-established proper scoring rule for probability forecasts [23]—based on their marginal contributions to the accuracy of the collective forecast.

While in principle joint-outcome markets for GHG concentration and temperatures solves the circularity problem, it is not obvious that they will work in practice. To provide useful information, concentrations and temperatures should be finely partitioned (e.g., 10ppm intervals for carbon dioxide, 0.1˚C for temperature). The resulting outcome space is an order of magnitude larger than those featured in previous successful trials [24]. It also cannot be assumed that market participants will be able to engage with a complex market structure in a way that allows their views to be effectively expressed and aggregated.

To test the viability of the concept, a series of markets with prediction horizons of months, rather than years, but with two-dimensional outcome spaces and similarly high levels of granularity were constructed. Instead of targeting GHG concentrations and global temperature these markets predicted monthly temperature and rainfall for the United Kingdom. These variables were chosen to provide an interesting prediction problem on seasonal time scales, and one that would attract participants with comparable backgrounds and expertise to those that would be desirable in longer-range climate markets.

Another attractive feature of seasonal forecasting for the purposes of this experiment is that there are several approaches to producing forecasts on horizons of a few months ahead. Ocean temperatures can be used as statistical predictors of seasonal temperature and rainfall while fully dynamical CGCMs can also be used [25]. Seasonal forecasting therefore provides an example where different types of information—different types of models and expertise about their relative merits—should be synthesized to produce a prediction.

## Market design

### Automated market maker

To test the viability of a granular joint-outcome market, six joint markets to predict the monthly averaged daily high temperature and total monthly rainfall for the United Kingdom, for the months of April to September 2018, were constructed. The UK Met Office publishes these monthly summary statistics. Temperature was partitioned into intervals of 0.2˚C, ranging from 0˚C to 25˚C, with open intervals covering temperatures below 0˚C and above 25˚C. Rainfall was partitioned into intervals of 5mm from zero to 200mm, with an additional interval covering totals above 200mm. This partitioning gave 127×41 = 5207 distinct outcomes, which were mutually exclusive and comprehensively exhaustive (see Fig 1).

To overcome potential liquidity issues with so many outcomes, an LMSR automated market maker was used [22]. This algorithm will always quote a price at which it will buy or sell individual outcomes or combinations thereof.

The LMSR market making algorithm uses the cost function

$$C(\boldsymbol{q}) = b \log \left( \sum_{i=1}^{m} e^{q_i/b} \right)$$

where $m$ is the number of outcomes, $q_i$ is the market maker's exposure to outcome $i$, and $b$ is a liquidity parameter that determines how much prices change in response to a given change in the maker maker's exposure. The parameter $b$ also determines the maximum net loss that the market maker can suffer. For the temperature-rainfall markets values of $m = 5207$ and $b = 500$ were used. Each participating team was endowed with 1000 credits which were transferable between the six markets. This transferability allowed participants to deploy their credits at the prediction horizons where they believed they had the greatest informational advantage.

The prevailing (marginal) prices the market maker assigns to each outcome, $p_i$, are given by the derivative of the cost function, that is

$$p_j = \frac{\partial C}{\partial q_j} = \frac{e^{q_j/b}}{\sum_{i=1}^{m} e^{q_i/b}}$$

Note that these marginal prices are normalized so that $\sum_{j=1}^{m} p_j = 1$, which is why the outcomes are defined to be mutually exclusive and comprehensively exhaustive. It is also why each $p_j$ can be formally interpreted as the probability of outcome $j$.

**Fig 1. Prices in the market for U.K. average daily high temperature and total monthly rainfall for July 2018 on 15th May 2018.** The grid shows the partitioning of the joint-outcome space into 5,207 separate outcomes, each covering a 0.2˚C interval in temperature and 5mm in rainfall. The extreme intervals for temperature and the maximum interval for rainfall are open intervals. The black dots show values of temperature and rainfall for each July since 1908.

If an order changes the market maker's exposure from $q_i$ to $q_i+w_i$ (for $i = 1,\cdots,m$) the market maker quotes an asking price given by

$$C(\boldsymbol{q} + \boldsymbol{w}) - C(\boldsymbol{q}) = b \log\left( \sum_{i=1}^{m} e^{(q_i+w_i)/_b} \right) - b \log\left( \sum_{i=1}^{m} e^{q_i/_b} \right)$$

An elegant feature of the LMSR marker maker is that the reward participants receive is linear in the logarithmic scoring rule [22]. This is a strictly proper scoring rule that, if the participants' utilities are linear in the number of credits they accumulate, will incentivize participants to reveal their true beliefs about the probabilities of each outcome [26]. That is, the LMSR is "incentive compatible".

To illustrate the relationship between the market maker and the logarithmic scoring rule, consider the situation where a single participant believes the probability of outcome $i$ occurring is $f_i$ ($i = 1,\cdots,m$). They will buy outcomes until the prices set by the market maker equal

their beliefs, that is until

$$f_j = \frac{e^{q_j/b}}{\sum_{i=1}^{m} e^{q_i/b}}$$

The exposure of the market maker that corresponds to these prices is given by

$$q_j = b \log f_j + b \log\left(\sum_{i=1}^{m} e^{q_i/b}\right) = b \log f_j + bA$$

where $A = \log\left(\sum_{i=1}^{m} e^{q_i/b}\right)$.

Buying this exposure will cost the participant

$$C_f = b \log(e^A \sum_{i=1}^{m} f_i) - b \log m = bA - b \log m$$

If outcome $k$ is the one that ultimately occurs, then the participant will receive $q_k$, so their net reward will be

$$q_k - C_f = b \log f_k + b \log m$$

which is linear in $\log f_k$, which is the logarithmic scoring rule. The largest net reward the participant can receive, and hence the largest net loss the market maker can suffer, is bounded by the case $f_k = 1$, and is given by $b \log m$. This shows how the parameter $b$ controls the market maker's potential loss.

The software platform allows participants to define contracts, $\mathbf{w} = (w_1, \cdots, w_m)$, consisting of combinations of outcomes in the joint-outcome space and then trade these contracts via the market maker. Since contracts can be defined to cover every possible interval in one variable, participants can take positions solely in temperature or rainfall. Participants can sell contracts they had previously bought at any time, although "shorting" (selling contracts that they do not own) is not permitted. However, because the pricing algorithm ensures the total price of the entire outcome space is always 1.00, if participants believe some outcomes are overpriced, they can exploit this by purchasing underpriced outcomes. Participants can trade using the platform's user interface (UI) or programmatically through an application programming interface (API). The UI only allows $w_i \in \{0,1\}$, whereas the API allows arbitrary non-negative values of $w_i$ to be used.

The LMSR market maker rewards participants according to same scheme proposed by Sandroni [13] when the proper scoring rule used is the logarithmic score, except that the "benchmark" probabilities which participants must improve upon are provided by the prevailing market prices rather than the forecast user.

## Initialization auction

If the market maker's initial exposure is $q_i = 0$ ($i = 1, \cdots, m$) then initial prices are $1/m$ for all outcomes. This is typically unrealistic and, as soon as the market opens, participants will rush to snap up underpriced outcomes. To try and avoid this rush, the initial prices in the markets were determined using an auction. For one week before market opening participants could submit orders consisting of a contract definition, $\mathbf{w} = (w_1, \cdots, w_m)$, the price the participant was prepared to pay per unit of that contract, $R$, and the maximum number of contracts they were prepared to buy at that price, $x_{max}$.

Once the bidding was closed, an algorithm prioritized orders based on the *premium* offered above what the market maker would ask for in live trading. The maximum premium for an

order may correspond to a fraction of that order, $x<x_{max}$, so $x$ had to be determined for each order.

The price of the contract after $x$ contracts are filled is

$$P(x) = \frac{\sum_{i=1}^{m} w_i e^{(q_i+xw_i)/b}}{\sum_{i=1}^{m} e^{(q_i+xw_i)/b}}$$

If $x$ contracts are filled, then the premium is

$$L(x) = \text{PRICE OFFERED} - \text{MARKET MAKER QUOTE}$$

$$L(x) = xR - \left\{ b\log\left(\sum_{i=1}^{m} e^{(q_i+xw_i)/b}\right) - b\log\left(\sum_{i=1}^{m} e^{q_i/b}\right) \right\}$$

The maximum obtainable premium was determined by solving

$$\frac{dL}{dx} = R - P(x) = 0$$

which shows that the order that maximizes the premium is the one that raises the price of the contract to the bid price.

When $P(0){\geq}R$ then the premium was negative. Such orders were not filled.

When $P(x_{max}){\leq}R$ then the maximum premium corresponded to the full order of $x_{max}$.

When $P(0)<R<P(x_{max})$ then the maximum premium was for a partial fulfillment of the order, $0<x<x_{max}$. The value of $x$ was determined with a line search.

Once all submitted orders had been ranked by the maximum obtainable premium, the one with the highest premium (if it was non-negative) was filled. Premiums were then recalculated and the remaining orders reranked. This process was repeated until there were no more unfilled orders with a non-negative premium.

Teams were recruited from British universities. There were 28 teams, drawn from departments with expertise in meteorology and climate science, as well as statistics, machine learning and economics. The invited teams were each endowed with 1000 credits with which to trade. The ten teams with the most credits after the September market had been settled received awards of £10,000 for first place, £9,000 for second, £8,000 for third, and so on down to £1,000 for tenth place. It should be acknowledged that, under these pay offs, the utility of participants is not linear in the number of credits they accumulate, undermining the properness of the scoring rule [27]. The gradual decline in rewards with placing was to mitigate the potential distortion introduced by the tournament element.

## Results

All six markets were opened on 12th March 2018 with auctions to set the initial prices. 23 orders from five teams were filled during the auction clearing process. Most teams did not trade until the markets opened for real-time trading a week later. Between then and 30th September 23,284 orders were filled across the six markets. Fig 1 shows the prices in the joint-outcome space for the July market on 15th May. The joint distribution captures the negative relationship between temperature and rainfall—hot summers tend to be dry—that is seen in the distribution of historic Julys. Early summer 2018 turned out to be very warm, with the warmest May on record and the second warmest June and July.

Fig 2 shows the daily number of trades and the daily trade volume (in credits) for each of the markets. Most trading in a market occurred during the month that the market was

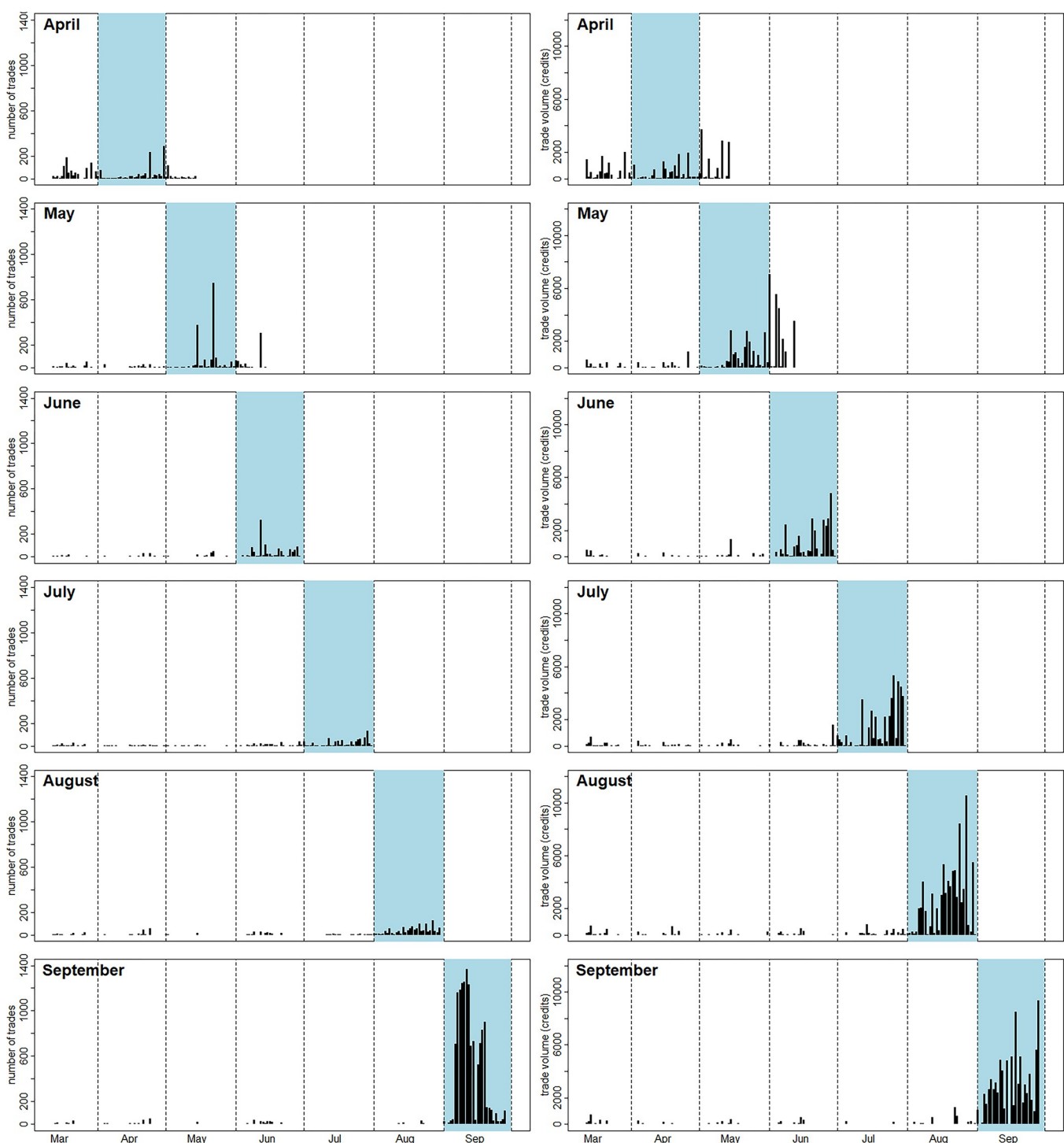

**Fig 2. The daily number of trades and volume for each of the monthly markets.** All the markets opened on 12th March 2018. The markets for April and May were closed for trading on 15th May and 15th June respectively, after the Met Office had published the official figures. The markets for June to September were closed on the last day of the relevant month, before publication of the official number.

predicting. This makes sense because seasonal forecasting for mid-latitudes is particularly challenging, compared with the tropics where the presence of El Niño-Southern Oscillation can allow skillful forecasts to be made months in advance. Once the target month has started, however, the flow of information to participants, in the form of medium-range weather forecasts and observations, increased and they injected this information into the market. September stands out due to the high number of trades—sometimes more than 1,000 in a day—although the overall volume of trades (in terms of credits) was not as exceptional. The average trade size fell to 6.7 credits in September, compared with 50.7 credits in August. Most of these September trades were made programmatically through the platform's API.

Fig 3 shows how the marginal distributions of prices for temperature and rainfall evolved throughout the duration of the markets. The April and May markets were held open until the 15th of the following month, shortly after the official numbers from the Met Office were available. This was done to explore the behaviour of the markets during the period after the temperature and rainfall values were, in principle, knowable, but before publication of the official values for settlement. Some trading did occur in this period, but it did not affect the expected values implied by the market. Subsequent markets were closed on the last day of the month they referred to, before publication of the official numbers. The decision to close later markets on the last day of the month was made to prevent participants being rewarded for trading on the publication of the official numbers. If the participants were rewarded proportionally to their credit accumulation such last minute trading would only be at the expense of the market maker but, because of the tournament-like structure of the rewards, participants who had made contributions to forecast accuracy earlier in the month could conceivably lose out to participants who only traded after the end of the month.

Temperature markets for June and July both started trending upwards during May, anticipating a continuation of that month's warm weather. The August and September markets showed a less pronounced upward trend during May, presumably because of the lower correlation between monthly temperatures at intervals beyond a couple of months. All the markets showed convergence to the actual values of temperature and rainfall as they approached the end of the relevant month.

## Discussion

The pilot markets clearly demonstrate the viability of large, two-dimensional outcome spaces when combined with an LMSR automated market maker. They also demonstrate that meteorologists and climate scientists—the types of experts who should participate in long-range climate prediction markets—can quickly adapt to prediction markets as a mechanism through which to contribute their expertise. Neither the prediction market institution itself, nor the complexities associated with the joint-outcome market format, posed an impediment to participation. The complexity did highlight the advantage of providing participants with access to an API. This allowed them to integrate their own quantitative models directly into trading, facilitating the synthesis of these models, and other tacit knowledge possessed by participants, into the collective forecasts of probabilities.

The results from the trial markets provide a proof-of-concept for using prediction markets to address the problems of aggregation, circularity, and accountability, in longer-range climate prediction. Joint outcome prediction markets for GHG concentrations and global temperatures could provide evolving estimates of both quantities as well as transient climate sensitivity. A probabilistic prediction of GHG concentrations over the next two or three decades would allow relative likelihoods to be assigned to emissions scenarios produced by organizations such as the IPCC, International Energy Agency (IEA) and the Network for Greening the

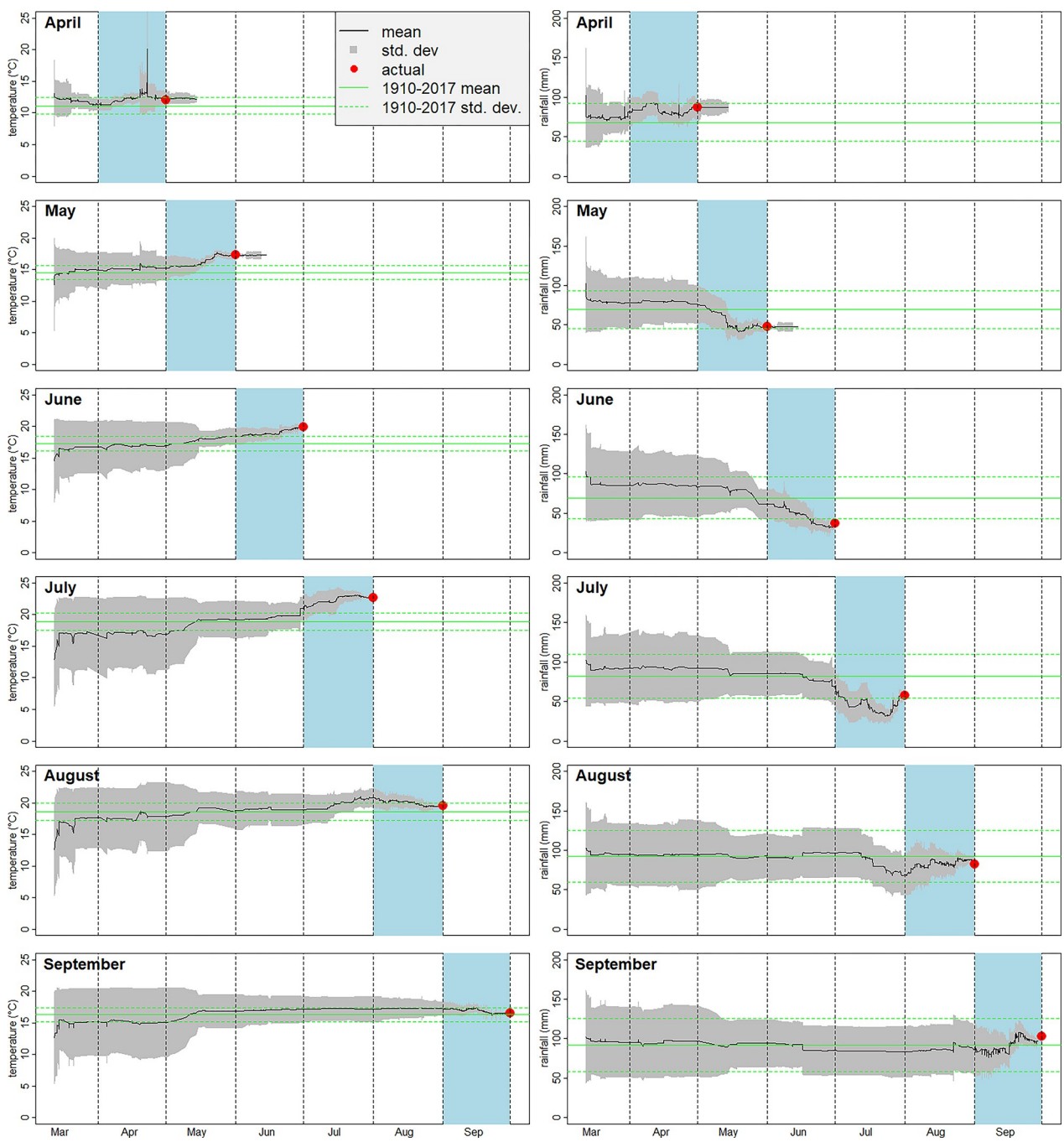

**Fig 3. The evolution of the marginal price distributions for UK average daily high temperature and monthly rainfall for the markets covering April to September 2018.** All the markets opened on 12th March 2018. The markets for April and May were closed for trading on 15th May and 15th June respectively, after the Met Office had published the official figures (the "actual" denoted by the red dots). The markets for June to September were closed on the last day of the relevant month, before publication of the official number. The green lines show the mean and standard deviation for the relevant month calculated using 1910–2017 data.

Financial System (NGFS). Likelihoods which are the product of the collective intelligence of diverse experts, could help anchor the debate surrounding scenarios both for policy making and the disclosure of climate risks, which is increasingly expected of organizations. Implied probabilities of emissions pathways would be a valuable reference for organisations wishing to

select scenarios for scenario analysis and "stress testing" exercises. These market-based likelihoods of different emissions pathways would also provide information about the credibility of policies to curb emissions. For example, if an international treaty to drastically reduce emissions was agreed but the market implied probabilities of lower emission scenarios did not increase, this could be interpreted as a sign of a lack of confidence in the effectiveness of the treaty.

As well as GHG concentrations and global temperatures, a platform for prediction markets which attracts expert participants could be used to produce collective forecasts of many climate-related variables. This could include those pertaining to physical risks, such as sea-level rise, and regional temperatures and rainfall, but also those that drive transition risks, such as the uptake of renewable technologies.

Establishing a prediction market to aggregate forward-looking climate information would remove the need to select a single forecast provider, which is a challenge for users of climate forecasts because, unlike with weather forecasts, they cannot rely on the forecaster's track record to evaluate their quality. Because the aim of the markets would be to obtain information from participants, rather than revenue, these participants would not have to "pay-to-play", avoiding the regulatory issues that have hindered the development of prediction markets. Instead, they could be endowed with credits representing a share of funding provided by organizations seeking credible, collective forecasts of climate risks. Some of this funding would also be used to provide the subsidy required by the market maker to incentivize participants. The markets could provide a method for distributing funding to researchers in climate-related areas in a way that makes them accountable for the accuracy of forecasts and allows the providers of funding to specify the outcomes for which they seek predictions.

An objection to using prediction markets to inform climate change policies is that some participants may attempt to manipulate the markets to distort climate predictions to either exaggerate or downplay the risks of climate change, depending on the agenda of the participant. This is more of a risk with open "pay-to-play" markets, where a participant willing to lose money can distort the forecast generated by the market. Such behaviour, however, will create profit opportunities for other participants, and these additional incentives can actually enhance accuracy [28]. Nevertheless, the suspicion by decision makers that manipulators are taking part in a prediction market can lead them to under-utilize the information generated by the market, even if there are no manipulators present [29]. It is therefore crucial that prediction markets intended for use by policy makers are monitored so forecast users are confident they are not being manipulated.

## Conclusion

The experimental markets described in this paper prove that the problem of limited liquidity, that has hindered previous prediction markets for climate-related variables, can be overcome with a subsidised market maker. They also demonstrate that using an automated market maker allows markets with large numbers of outcomes, including joint outcomes, to function.

Although the markets converged well to the actual values, six independent forecasts are not sufficient to provide a statistically meaningful determination of the accuracy of the prediction market approach compared with other methods. Furthermore, none of the markets had a forecast horizon of more than seven months, so whether markets with significantly longer horizons would behave in a similar way is an open question. Running markets with much shorter horizons would produce enough data to perform a statistically rigorous verification study, but shorter horizons would accentuate the question of the validity of the results to multi-year predictions. This validation problem is, of course, intrinsic to all methods for predicting long-

range climate, and lies at the heart of the information asymmetry discussed in the introduction. Therefore, the accountability provided by incentive-compatible compensation is a desirable feature for all forward-looking information. The markets described in this paper have since been included in a wider verification study of climate-related prediction market forecasts with horizons of up to one year [30]. However, the only way to test whether prediction markets can be effective for forecasting problems on horizons of years to decades is to establish markets on these time scales. These time scales are longer than the horizons of prediction markets that have been run up until now, so aspects of their design would require careful consideration: participants would need confidence in the stability and longevity of the institution operating the market funds backing the credits would have to be segregated, and the monetary value of credits should appreciate to reflect interest accrued by the segregated funds.

This study has demonstrated the ability of carefully designed prediction markets to efficiently aggregate expertise relevant to climate. Extending the concept to longer horizons could produce more credible forecasts of greater use for policy making, strategic planning, and climate-related risk disclosures.

## Acknowledgments

The authors would like to thank Winton Group, as well as participants in a 2016 workshop hosted by Winton Group which elaborated the initial market design. They would also like to thank the university teams who participated in the 2018 temperature-rainfall markets.

## Author Contributions

**Conceptualization:** Mark S. Roulston, Kim Kaivanto.

**Formal analysis:** Mark S. Roulston.

**Methodology:** Mark S. Roulston.

**Writing – original draft:** Mark S. Roulston, Kim Kaivanto.

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
