## [Decision Letter · Decision Letter 0]

6 Jul 2023

PONE-D-23-08410Joint-outcome Prediction Markets for Climate RisksPLOS ONE

Dear Dr. Roulston,

Thank you for submitting your manuscript to PLOS ONE. After careful consideration, we feel that it has merit but does not fully meet PLOS ONE’s publication criteria as it currently stands. Therefore, we invite you to submit a revised version of the manuscript that addresses the points raised during the review process.

We look forward to receiving your revised manuscript.

Kind regards,

Robin Haunschild

Academic Editor

PLOS ONE

Reviewers' comments:

Reviewer's Responses to Questions

**Comments to the Author**

1. Is the manuscript technically sound, and do the data support the conclusions?

Reviewer #1: Yes

Reviewer #2: No

2. Has the statistical analysis been performed appropriately and rigorously? 

Reviewer #1: Yes

Reviewer #2: No

3. Have the authors made all data underlying the findings in their manuscript fully available?

Reviewer #1: No

Reviewer #2: No

4. Is the manuscript presented in an intelligible fashion and written in standard English?

Reviewer #1: Yes

Reviewer #2: No

5. Review Comments to the Author

Reviewer #1: Predicting future climate is still very important in human history. This study proposes a joint-outcome-market approach to predict the monthly averaged daily high temperature and total monthly rainfall for the United Kingdom, for the months of April to September 2018. The approach has important contribution if the empirical result is more robust. However, the authors just adopt six joint markets and special period to support their claims. The study should give a clear conclusion instead of discussion.

Reviewer #2: Lack of Context: Without setting the stage, the work quickly dives into a discussion of climate forecasting. It would be useful to begin with some background information on the significance and present methods of climate prediction, as well as some discussion of the shortcomings of these methods. This would lay the groundwork for learning about the importance of non-traditional approaches like prediction markets.

The document alludes to the "circularity" (or interdependence) between climate projections and mitigation programs, but fails to provide a clear explanation of this connection. The importance of explaining why climate predictions are dependent on greenhouse gas emissions and how these projections guide efforts to reduce such emissions cannot be overstated. Readers may struggle to grasp the implications of this circular dependence without a thorough explanation.

Although the practicality of prediction markets with joint-outcome spaces is briefly discussed with reference to UK rainfall and temperature instances, the book does not give a thorough examination or analysis of the outcomes. To prove the usefulness of prediction markets in meeting the aforementioned issues, it would be helpful to incorporate in-depth data, statistical analysis, and comparisons with other prediction approaches.

The manuscript alludes to knowledge asymmetry as a challenge, but does not address the issue head-on or offer any answers. To assure the reliability of predictions, it is essential to investigate ways to deal with the information asymmetry that plagues prediction markets. The manuscript's argument might benefit from some illumination on this point.

The manuscript's narrow focus on monthly rainfall and temperature in the UK makes it difficult to evaluate the wider relevance of prediction markets for climate-related risks. Prediction markets could be useful for making global or regional climate forecasts, but there are some caveats and obstacles to be aware of. The manuscript's authority would also increase if it addressed potential objections to the use of prediction markets.

The paper only briefly touches on the use of prediction markets for policy making, planning, and risk disclosure, but it does not dive into the precise consequences or potential benefits of doing so. The manuscript's effect and relevance would increase if it included a more in-depth examination of how prediction markets might influence policy decisions, strengthen planning tactics, and boost risk communication.

6. PLOS authors have the option to publish the peer review history of their article (what does this mean?). If published, this will include your full peer review and any attached files.

Reviewer #1: No

Reviewer #2: No

---

## [Author Response · Author response to Decision Letter 0]

13 Jul 2023

Dear Dr. Haunschild,

We would like to submit our revised manuscript, “Joint-outcome Prediction Markets for Climate Risks” to be considered for publication in PLOS ONE as a Research Article.

We believe we have addressed the comments made by both reviewers. The changes we have made to our manuscript are described below.

3. Have the authors made all data underlying the findings in their manuscript fully available?

Reviewer #1: No

Reviewer #2: No

RESPONSE: The code used to produce the plots, and the underlying data (in two different formats) can be downloaded from https://drive.google.com/drive/folders/1oAC5LxyospVf00u-lpVlg04wtQjNqEYz

5. Review Comments to the Author

Reviewer #1: Predicting future climate is still very important in human history. This study proposes a joint-outcome-market approach to predict the monthly averaged daily high temperature and total monthly rainfall for the United Kingdom, for the months of April to September 2018. The approach has important contribution if the empirical result is more robust. However, the authors just adopt six joint markets and special period to support their claims. The study should give a clear conclusion instead of discussion.

RESPONSE: We have added a conclusion section [L459-L486] which makes clear that we are claiming that the markets demonstrate that using an LMSR market maker overcomes the liquidity issues experienced by previous climate-related markets, and allow a joint-outcome market, with a large number of outcomes to actually work. As Reviewer #1 points out, there are only six markets, and we mention that this is not enough to make conclusions about the statistical accuracy of the prediction market approach compared to other methods. This, however, is an intrinsic problem with all climate forecasting because of the horizons involved. 

Reviewer #2: Lack of Context: Without setting the stage, the work quickly dives into a discussion of climate forecasting. It would be useful to begin with some background information on the significance and present methods of climate prediction, as well as some discussion of the shortcomings of these methods. This would lay the groundwork for learning about the importance of non-traditional approaches like prediction markets.

RESPONSE: The introduction now includes a discussion of current approaches to climate forecasting using coupled general circulation models (CGCMs) [L47-L68] as well as an explanation of “downscaling” these models to produce higher resolution forecasts [L96-L110]. We also emphasize that prediction markets are not a replacement for existing methods for climate forecasting but a mechanism to synthesize different approaches; both traditional CGCMs and non-traditional methods such as ML and AI [L145-L148]. 

The document alludes to the "circularity" (or interdependence) between climate projections and mitigation programs, but fails to provide a clear explanation of this connection. The importance of explaining why climate predictions are dependent on greenhouse gas emissions and how these projections guide efforts to reduce such emissions cannot be overstated. Readers may struggle to grasp the implications of this circular dependence without a thorough explanation.

RESPONSE: We have expanded the explanation of circularity and provided a description of how CGCM forecasts are contingent on the GHG concentration scenarios used [L47-L68].

Although the practicality of prediction markets with joint-outcome spaces is briefly discussed with reference to UK rainfall and temperature instances, the book does not give a thorough examination or analysis of the outcomes. To prove the usefulness of prediction markets in meeting the aforementioned issues, it would be helpful to incorporate in-depth data, statistical analysis, and comparisons with other prediction approaches.

RESPONSE: Unfortunately, the data set essentially consists of only six independent forecasts, which severely constrains what conclusions can be drawn concerning accuracy. This is, of course, an intrinsic issue with making any forecasts on seasonal time scales and even more so for forecasts on climate time scales. We discuss this issue in the manuscript [L457-L465]. All the pricing data for the market is available for download so any reader who wishes to conduct a specific form of statistical analysis is free to do so. Our primary conclusion is that using subsidised AMMs overcomes the liquidity problem encountered in previous markets for climate-related outcomes, and with a large number of outcomes, and therefore such AMMs make highly granular joint-outcome markets viable. 

The manuscript alludes to knowledge asymmetry as a challenge, but does not address the issue head-on or offer any answers. To assure the reliability of predictions, it is essential to investigate ways to deal with the information asymmetry that plagues prediction markets. The manuscript's argument might benefit from some illumination on this point.

RESPONSE: Asymmetric information is one of the key problems that prediction markets resolve, through what is termed ‘incentive compatibility’ – i.e., given the way in which a forecaster’s payoff in a prediction market is dependent on the provision of accurate forecasts, the forecaster has the incentive to truthfully provide accurate forecasts. In one sense therefore, prediction markets are an antidote to the problem of asymmetric information, rather than yet another context where information asymmetry is a problem. 

We have included more information about the problem of information asymmetry, including a reference (Sandroni 2014) that explains how paying forecasters according to a proper scoring rule is incentive compatible – even in the case of a single forecast [L128-L130].

The manuscript's narrow focus on monthly rainfall and temperature in the UK makes it difficult to evaluate the wider relevance of prediction markets for climate-related risks. Prediction markets could be useful for making global or regional climate forecasts, but there are some caveats and obstacles to be aware of. The manuscript's authority would also increase if it addressed potential objections to the use of prediction markets.

RESPONSE: There is now a discussion of the possibility of manipulation of prediction markets, and how this can undermine the confidence decision makers have in the information they produce, even when its accuracy is not compromised [L439-L449].

The paper only briefly touches on the use of prediction markets for policy making, planning, and risk disclosure, but it does not dive into the precise consequences or potential benefits of doing so. The manuscript's effect and relevance would increase if it included a more in-depth examination of how prediction markets might influence policy decisions, strengthen planning tactics, and boost risk communication.

RESPONSE: In the discussion we now talk about how the output of prediction markets might be useful to policy makers and useful for communicating information about climate risks [L408-L420].

Yours sincerely

Mark Roulston

---

## [Decision Letter · Decision Letter 1]

7 Aug 2024

Joint-outcome Prediction Markets for Climate Risks

PONE-D-23-08410R1

Dear Dr. Roulston,

We’re pleased to inform you that your manuscript has been judged scientifically suitable for publication and will be formally accepted for publication once it meets all outstanding technical requirements.

Kind regards,

Robin Haunschild

Academic Editor

PLOS ONE

Additional Editor Comments (optional):

Two reviewers recommended to accept the manuscript. The reviewer who recommended to reject the paper did not provide comments regarding the evaluation policy of PLOS ONE (i.e., scientific rigor).

Reviewers' comments:

Reviewer's Responses to Questions

**Comments to the Author**

1. If the authors have adequately addressed your comments raised in a previous round of review and you feel that this manuscript is now acceptable for publication, you may indicate that here to bypass the “Comments to the Author” section, enter your conflict of interest statement in the “Confidential to Editor” section, and submit your "Accept" recommendation.

Reviewer #3: (No Response)

Reviewer #4: All comments have been addressed

Reviewer #5: All comments have been addressed

2. Is the manuscript technically sound, and do the data support the conclusions?

Reviewer #3: Partly

Reviewer #4: Yes

Reviewer #5: Yes

3. Has the statistical analysis been performed appropriately and rigorously? 

Reviewer #3: Yes

Reviewer #4: Yes

Reviewer #5: Yes

4. Have the authors made all data underlying the findings in their manuscript fully available?

Reviewer #3: No

Reviewer #4: Yes

Reviewer #5: Yes

5. Is the manuscript presented in an intelligible fashion and written in standard English?

Reviewer #3: Yes

Reviewer #4: Yes

Reviewer #5: Yes

6. Review Comments to the Author

Reviewer #3: The sample time of the articles is too old, the contribution is very limited, and the comments of the reviewers are not completed.

Reviewer #4: After the revisions made suggested by other reviewers, the paper can be acepted.

The paper is well structured.

Reviewer #5: (No Response)

7. PLOS authors have the option to publish the peer review history of their article (what does this mean?). If published, this will include your full peer review and any attached files.

Reviewer #3: No

Reviewer #4: **Yes: **Hélder Silva Lopes

Reviewer #5: No

---

## [Editor Report · Acceptance letter]

12 Aug 2024

PONE-D-23-08410R1 

PLOS ONE

Dear Dr. Roulston, 

I'm pleased to inform you that your manuscript has been deemed suitable for publication in PLOS ONE. Congratulations! Your manuscript is now being handed over to our production team.

Kind regards, 

on behalf of

Dr. Robin Haunschild 

Academic Editor

PLOS ONE